# Evidence for the Presence of *Borrelia burgdorferi* Biofilm in Infected Mouse Heart Tissues

**DOI:** 10.3390/microorganisms12091766

**Published:** 2024-08-26

**Authors:** Sahaja Thippani, Niraj Jatin Patel, Jasmine Jathan, Kate Filush, Kayla M. Socarras, Jessica DiLorenzo, Kunthavai Balasubramanian, Khusali Gupta, Geneve Ortiz Aleman, Jay M. Pandya, Venkata V. Kavitapu, Daina Zeng, Jennifer C. Miller, Eva Sapi

**Affiliations:** 1Lyme Disease Research Group, Department of Biology and Environmental Science, University of New Haven, 300 Boston Post Road, West Haven, CT 06516, USA; sthip3@unh.newhaven.edu (S.T.); npate42@unh.newhaven.edu (N.J.P.); jjath1@unh.newhaven.edu (J.J.); katherine.r.filush@gmail.com (K.F.); kmsocarras@gmail.com (K.M.S.); jdilo2@unh.newhaven.edu (J.D.); kunthavai.89@gmail.com (K.B.); khusali.gupta@umassmed.edu (K.G.); gorti2@unh.newhaven.edu (G.O.A.); jpand2@unh.newhaven.edu (J.M.P.); vkavi2@unh.newhaven.edu (V.V.K.); 2Department of Biological Sciences, North Carolina State University, 3510 Thomas Hall, 112 Derieux Pl, Raleigh, NC 27607, USA; daina.zeng@gmail.com (D.Z.); jcmille4@gmail.com (J.C.M.)

**Keywords:** biofilm, carditis, inflammation, immunohistochemistry, atomic force microscopy, C-reactive protein

## Abstract

*Borrelia burgdorferi*, the bacterium responsible for Lyme disease, has been shown to form antimicrobial-tolerant biofilms, which protect it from unfavorable conditions. Bacterial biofilms are known to significantly contribute to severe inflammation, such as carditis, a common manifestation of Lyme disease. However, the role of *B. burgdorferi* biofilms in the development of Lyme carditis has not been thoroughly investigated due to the absence of an appropriate model system. In this study, we examined heart tissues from mice infected with *B. burgdorferi* for the presence of biofilms and inflammatory markers using immunohistochemistry (IHC), combined fluorescence in situ hybridization FISH/IHC, 3D microscopy, and atomic force microscopy techniques. Our results reveal that *B. burgdorferi* spirochetes form aggregates with a known biofilm marker (alginate) in mouse heart tissues. Furthermore, these biofilms induce inflammation, as indicated by elevated levels of murine C-reactive protein near the biofilms. This research provides evidence that *B. burgdorferi* can form biofilms in mouse heart tissue and trigger inflammatory processes, suggesting that the mouse model is a valuable tool for future studies on *B. burgdorferi* biofilms.

## 1. Introduction

Lyme borreliosis, a tick-borne multisystemic illness caused by the spirochete *Borrelia burgdorferi*, has grown into a major public health concern [1,2,3]. The frontline treatment for Lyme disease is the administration of antibiotics [3]. However, relapse often occurs when the treatment is discontinued, irrespective of the antibiotic used [4,5,6]. This recurrence may be attributed to various factors, including the survival of *B. burgdorferi,* as suggested by multiple in vivo studies which indicated that *B. burgdorferi* spirochetes can persist after antimicrobial treatment though they remain uncultivable [6,7,8].

It has been proposed that these antibiotic-tolerant spirochetes might result from the transformation of *B. burgdorferi* into alternative morphological forms, such as “round bodies (cysts and granules)”, a concept first identified and studied in the 1990s by Dr. O. Brorson and Dr. A. MacDonald [9,10,11,12,13]. Our research group discovered that *B. burgdorferi* can also form biofilms, a known bacterial structure formed as a stress response to exposure to antibiotics or any adverse environmental condition [14]. In other bacteria, such as *Staphylococcus aureus*, *Streptococcus*, and *Salmonella* spp., biofilms have been identified in several chronic diseases including periodontitis, osteomyelitis, and chronic lung infection in cystic fibrosis patients [15,16,17,18,19]. These biofilms are associated with chronic conditions and are challenging to eradicate due to their significantly increased antibiotic resistance, which can be up to 1000 times greater than that of their free-living counterparts [20]. The resistance of biofilms is attributed to multiple mechanisms, such as the incomplete penetration of antibiotics into the biofilm matrix, antibiotic inactivation due to the altered microenvironment within the biofilm, and the presence of bacterial persisters [21,22].

In our previous studies, we provided both in vitro and in vivo evidence that *B. burgdorferi* biofilm exists within infected human tissues [14,23]. Many key biofilm traits, such as structural rearrangements and the development of protective matrices on the surface, were identified [14]. Our atomic force microscopy data revealed the formation of channels and protrusions during biofilm development [14]. Specific biofilm markers, including protective layers containing alginate, were found in *B. burgdorferi* biofilms [14,23]. Alginate, a key component of protective matrices, can be used to demonstrate that these aggregates are true biofilms [24]. We also demonstrated that *B. burgdorferi* biofilms are highly tolerant to Lyme disease antibiotics [25], suggesting that antibiotics alone are insufficient for the removal of these biofilms.

Our follow-up studies also demonstrated that *B. burgdorferi* biofilms are present in infected human tissues and can trigger a chronic inflammatory response. This was observed in a 53-year-old female patient who had been undergoing extensive antibiotic treatment for a 16-year-long illness [26]. The study confirmed the presence of CD3+ infiltrating T lymphocytes in the vicinity of *B. burgdorferi/*alginate-positive biofilm structures in brain, liver, kidney, and heart tissues [26]. Using confocal microscopy, our data provided further evidence of *B. burgdorferi* biofilms deeply embedded in the heart tissues alongside individual spirochetes [26].

A well-documented inflammatory complication associated with Lyme disease is endocarditis, reported to occur in approximately ~10% of cases within weeks to months of the initial infection [27]. Lyme carditis is defined as the presence of a high-degree atrioventricular block accompanying a Lyme disease diagnosis and can be fatal if untreated [28,29]. While bacterial biofilms are widely reported to be a significant factor in the development of carditis and its antibiotic resistance [30], the potential role of *B. burgdorferi* biofilm in the development of Lyme carditis has not yet been studied.

While we have evidence that *B. burgdorferi* biofilms exist in vivo in human heart tissues [26], we cannot use those tissues for more comprehensive studies. Therefore, we need a model system to study both the formation of biofilms and their effect on those tissues. Mouse models have been widely used to investigate the mechanism of *B. burgdorferi* infection in mammals [31,32,33,34]. However, none of these previous studies focused on finding biofilms in infected heart tissues. The only evidence that *B. burgdorferi* might form biofilms in murine tissues came from an ex vivo study in which murine skin biopsy samples were infected with *B. burgdorferi* [35]. Hence, it could be beneficial to evaluate whether *B. burgdorferi* can form biofilms in infected murine heart tissues and to assess their effect on host inflammatory responses.

In summary, this study focused on investigating biofilm formation and inflammatory responses in archived *B. burgdorferi*-infected mouse tissues with the goal of determining whether this formation indeed provides refuge for *B. burgdorferi* in a murine model. For these experiments, *B. burgdorferi* and biofilm-specific immunohistochemistry (IHC) and combined FISH/IHC methods with different microscopy techniques were used to find biofilms and the heart-associated inflammatory marker, C-reactive protein. By focusing on infected mouse heart tissues and investigating the presence of *B. burgdorferi* in relation to biofilm formation, this research aims to elucidate the underlying mechanisms of Lyme carditis and contribute to the development of effective diagnostic and treatment strategies.

## 2. Materials and Methods

### 2.1. Infection of Mice with B. burgdorferi

For this study, 4–6-week-old in-bred C3H/HeN mice were purchased from Charles River Laboratories (Wilmington, MA, USA) and housed under standard conditions of temperature, nutrition, and light in the animal facility of the Department of Laboratory Animal Resources at North Carolina State University. The midline of the back of the mice was injected with a low passage of *B. burgdorferi* N40 strains (20 mice in total in 2 groups). Four uninfected control mice received an equal volume of BSK-II growth medium as a placebo. All mice were euthanized at 4 weeks post-infection. Blood for serum was collected post-mortem which was used for ELISA analyses to measure *B. burgdorferi*-specific mouse IgG antibodies, as confirmation of successful infection. Additionally, 2–3 mm biopsies of skin (at the injection site), ears, the heart, and the rear ankle joints were aseptically excised and cultivated in BSK-II complete medium to further confirm *B. burgdorferi* infection. For immunohistochemical and FISH studies, the tissues were fixed using 4% formalin for 48 h at 4 °C, dehydrated in ethanol, washed in xylene, embedded in paraffin blocks, and sectioned for this study by McClain Laboratories LLC (Smithtown, NY, USA). All immunohistochemical, FISH, and microscopy studies were performed at the University of New Haven.

All in vivo experimental manipulations described above have been approved by the NCSU IACUC (protocol # 13-066-B), and by the NCSU Biosafety Committee (protocol # 2013-03-403r). The University of New Haven IACUC granted an exception for the use of the archived tissues.

### 2.2. DNA Extraction/PCR

DNA was extracted from N40-infected and uninfected C3H mouse tissues as described elsewhere [36] using sequential digestions with a 0.1% collagenase A solution and a 0.2 mg/mL proteinase K solution (Sigma Aldrich, St. Louis, MO, USA). Following a series of phenol/chloroform extraction and ethanol precipitation steps, the purified DNA was diluted to 5 µg/mL for use in qPCR assays. A qPCR was conducted using iQ SYBR Green Supermix following the manufacturer’s instruction (Bio-Rad, Hercules, CA, USA).

The target genes analyzed were the *B. burgdorferi* RecA nTM17.F (5′-GTG GAT CTA TTG TAT TAG ATG AGG CTC TCG-3′) and nTM17.R (5′-GCC AAA GTT CTG CAA CAT TAA CAC CTA AAG-3′) and the mouse *nidogen* were nido. F (5′-CCA GCC ACA GAA TAC CAT CC-3′) and nido. R (5′-GGA CAT ACT CTG CTG CCA TC-3′). The PCR reactions were conducted using the MyiQ2 Two-color Real-time PCR Detection System (Bio-Rad, Hercules, CA, USA) and the reaction conditions were as follows: initial denaturation at 95 °C for 3 min, 40 cycles of 95 °C/15 s; then one cycle of 95 °C/1 min, 60 °C/1 min, and 71 cycles of 60–90 °C/10 s.

### 2.3. Immunohistochemistry (IHC)

The paraffin-embedded, formalin-fixed tissue sections (4 µm) were deparaffinized on a slide warmer for 15 min followed by three 5-min xylene washes. The tissues were rehydrated in a series of ethanol washes for 3 min each with decreasing concentrations of 100%, 100%, 90%, and 70%. This step was followed by a 25-min wash under slow-running tap water to remove the traces of ethanol from the slide.

The water bath was equilibrated to 99 °C and slides were immersed into a Coplin jar with preheated mediated antigen retrieval buffer (10 mM sodium citrate, 0.05% Tween 20, pH 6.0) for 10 min. The Coplin jar was then removed from the water bath and set at room temperature for 20 min to cool the buffer. This step was followed by placing the slides under slow-running tap water to remove the traces of the sodium citrate buffer for 10 min. Before proceeding with immunostaining, the slides were rinsed with 1× phosphate buffered saline (PBS, Sigma Aldrich, St. Louis, MO, USA) mixed with 1% bovine serum albumin solution (BSA, Sigma Aldrich, St. Louis, MO, USA) and distilled water for 5 min each. The sections were then blocked with a 1:200 dilution of goat serum (Thermo Scientific, Waltham, MA, USA) in 1× PBS for 1 h in a humidified chamber at room temperature (RT). Excess solution was gently removed using a Kimwipe and the slides were rinsed with 1× PBS + 1% BSA and distilled water, each for 5 min. The slides were then treated with a polyclonal anti-alginate antibody made in rabbit (generously provided by Dr. Gerald Pier, Harvard Medical School) which was diluted in a 1:200 ratio with 1× PBS and added to the slides. The slides were incubated at 4 °C overnight in a humidified chamber. The next day the slides were gently rinsed with 1× PBS + 1% BSA and distilled water, each for 5 min. The tissue sections were then treated with a 1:200 dilution of the secondary anti-rabbit antibody with a fluorescent red tag (goat anti-rabbit IgG (H + L), DyLight 594 conjugated) and incubated for an hour at RT. The excess solution around the tissue was gently wiped away and the slides were washed as described above. A separate set of heart tissue slides was also stained for C-reactive protein rabbit mAb (Catalog No. A19003, ABclonal, Woburn, MA, USA) as described above and then treated with a 1:200 diluted secondary anti-rabbit antibody with a fluorescent red tag (goat anti-rabbit IgG (H + L), DyLight 594 conjugated). These were then incubated for an hour at RT and washed as described above.

The slides were then treated with a dilution of 1:100 of fluorescently labeled *B. burgdorferi* polyclonal antibody, (Catalog No. PA1-73005, Invitrogen, Waltham, MA, USA) in 1× PBS and incubated for an hour in a humidified chamber at RT. The slide sections were then washed as described above and then counterstained with 0.1% Sudan black (Sigma Aldrich, St. Louis, MO, USA) for 20 min. The slides were washed one last time and then mounted with Prolong Diamond Antifade Mounting media with DAPI (ThermoFisher Scientific, Waltham, MA, USA). Images were taken and processed using a Nikon Eclipse 80i (Nikon, Tokyo, Japan) and Leica Thunder fluorescent microscope (Leica Microsystems, Wetzlar, Germany) at 100×, 200×, and 400× magnifications.

Additionally, as negative controls, ten uninfected mouse tissue sections were used along with commercially bought tissue slides for normal mouse heart sections (Cat# MoFPT016, TissueArray, Derwood, MD, USA), which were stained following the same procedure described above.

### 2.4. Combined Fluorescent In Situ Hybridization (FISH) and IHC

A FISH/IHC test was performed on paraffin-embedded mouse heart tissues. The tissue sections were deparaffinized on a slide warmer for 15 min followed by three 5-min xylene washes. The tissues were rehydrated in a series of ethanol washes for 3 min each with decreasing concentrations of 100%, 100%, 90%, and 70%. Sections were then placed in a sodium borohydride solution (0.1 μg in 10 mL, Fisher Scientific, Waltham, MA, USA) for 20 min on ice. The slides were fixed with 4% paraformaldehyde (PFA) for 15 min at RT. The sections were washed in 2× saline sodium citrate (SSC) buffer (3.0 M NaCl, 0.3 M Na Citrate, and dH_2_O) and then digested with pre-warmed proteinase K solution (20 μg/mL in 50 mM Tris) (AmericanBio, Canton, MA, USA) for 10 min at 37 °C. Tissues were then refixed in 4% PFA for 10 min at room temperature. The slides were then placed into a preheated denaturing solution (70% *v*/*v* formamide, 2× SSC, 0.1 mM ethylenediaminetetraacetic acid [EDTA]) and incubated for 5 min at 95 °C. The slides were refixed again with 4% PFA for 10 min, washed, and added to the denaturing solution again for 2 min at 60 °C. The slides were washed in 2× SSC for 5 min at RT. Pre-hybridization followed for 4 h in hybridization buffer (50% *v*/*v* formamide, 10% *w*/*v* dextran sulfate [Sigma Aldrich, St Louis, MO, USA], 1% *v*/*v* Triton X−100 [Sigma Aldrich, St Louis, MO, USA], 2× SSC, and 2.5 ng of salmon sperm DNA (ThermoFisher Scientific, Waltham, MA, USA). The slides were incubated with a previously published 16S rDNA *Borrelia burgdorferi* FISH probe (26, FAM-5′-GGA TAT AGT TAG AGA TAA TTA TTC CCC GTT TG-3′) (Eurofins MWG Operon, Lancaster, PA, USA) at 48 °C for 18 h in the dark. A cover slip was placed on each slide to ensure hydration of the tissue throughout the incubation.

Post-hybridization, slides were washed twice in 2× SSC at RT for 5 min, and then washed in 0.2× SSC for 5 min at RT in the dark. After the 0.2× SSC wash in the FISH protocol, the sections were blocked with a 1:200 dilution of goat serum (Thermo Scientific, Waltham, MA, USA) for an hour at RT in a humidified chamber. The slides were washed 5× with PBS prior to the addition of the primary polyclonal anti-alginate antibody for overnight incubation at RT. The next day the slides were tagged with a 1:200 dilution of the secondary anti-rabbit antibody with a fluorescent blue tag (goat anti-rabbit IgG (H + L), DyLight 405 conjugated) and incubated for an hour at RT. This was then followed by a counterstaining step using 0.1% Sudan black for 20 min followed by several washes in 0.2× SSC. The slides were then mounted with PermaFluor mounting media (Thermo Scientific, Waltham, MA, USA) and stored at 4 °C. Images were taken using a Leica DM2500 fluorescent microscope (Leica Microsystems, Wetzlar, Germany) at 400× magnification. All FISH steps were repeated with several negative controls such as the following: (1) 100 ng random oligonucleotide, (5′-FAM-GCA TAG CTC TAT GAC TCT ATA CTG GTA CGT AG-3′), (2) 200 ng of unlabeled competing oligonucleotide added before the hybridization step [competing *Borrelia* (5′-CAA ACG GGG AAT AAT TAT CTC TAA CTA TAT CC-3′)] and a DNase treatment of the sections before the hybridization step to digest all genomic DNA (100 μg/mL for 60 min at 37 °C).

### 2.5. Atomic Force Microscopy (AFM)

To visualize the morphology of *B. burgdorferi* biofilms identified in the infected murine heart tissues, contact mode AFM imaging was performed on a Nanosurf Easyscan 2 AFM (Nanosurf, Liestal, Switzerland) using SHOCONG probes (Nanosensors™, Neuchatel, Switzerland). Images were processed using Gwyddion software, version 2.66 (http://gwyddion.net/) (accessed on 15 June 2024).

## 3. Results

### 3.1. B. burgdorferi Spirochete and Biofilm Presence in Mouse Heart Tissues

To evaluate whether *B. burgdorferi* forms biofilms upon experimental infection in a mouse model, 4–6-week-old C3H/HeN mice were injected with low passage *B. burgdorferi* N40 strain and allowed to develop Lyme disease and arthritis for 30 days. Various tissues and blood samples were collected for further analysis (Figure 1). To confirm successful infections and developing arthritis, the ankle joints were measured at 0 and 30 days of infection. Additionally, part of the tissue samples was placed into BSK-II complete media to evaluate the presence of live spirochetes. *B. burgdorferi*-specific ELISA for IgG antigen levels from blood samples and Real-Time PCR analyses for *B. burgdorferi* DNA presence were also performed on the collected tissues. Figure 1 shows the results of ELISA and the ankle joint measurements, revealing a significant increase in IgG levels and ankle sizes in all infected samples compared to the uninfected control. Additionally, results showed that live spirochetes can be cultured and *B. burgdorferi* DNAs can be detected in all infected tissues but not in the uninfected ones (Figure 1).

Next, we evaluated our primary antibodies for potential non-specific binding to any antigens in mouse heart tissues using immunohistochemical (IHC) methods. We purchased commercially available normal mouse heart tissue sections (10 in total) and stained them with *B. burgdorferi* and alginate antibodies using IHC as described in the Methods section. None of the heart tissue sections showed any detectable signal (Appendix A), confirming that the IHC method is specific and does not produce any background staining in mouse heart tissues. Consequently, we proceeded to investigate the different morphological forms of *B. burgdorferi* in infected (200 sections) and uninfected (50 sections) mouse heart tissues using the same *B. burgdorferi* and alginate (biofilm marker) specific IHC methods.

In Figure 2, representative images of infected tissues with uninfected negative controls are shown. The presence of *B. burgdorferi* spirochetes and aggregates are indicated in green within these tissues (Figure 2, Panels A, F, K). Concurrently, the utilization of an anti-alginate antibody resulted in the detection of *B. burgdorferi* biofilms (Figure 2, Panels B, G, L, red signal). In order to see cellular content, nuclear DNA was stained using DAPI (Figure 2, Panels D, I, N, S; blue color). The morphology of all the tissues was visualized using differential interference contrast microscopy (DIC) which showed the structure of the biofilms and how they are embedded in these tissues (Figure 2, Panels E, J, O, T). No *B. burgdorferi* or alginate staining were detected in any of the 50 uninfected heart tissue sections (Figure 2, Panels P, Q). Quantitative analyses showed that infected heart tissues had an average of 35 spirochetes and 0–6 biofilms per section. The biofilm sizes ranged from 20–200 μm. Appendix A provide additional IHC images for the presence of *B. burgdorferi* biofilms in uninfected and infected mouse heart tissues.

### 3.2. Combined FISH and IHC Confirmed the In Vivo Existence of B. burgdorferi Biofilm in Infected Mouse Heart Tissues

A previously published combined FISH/IHC method was utilized to confirm the presence of *B. burgdorferi* biofilm heart tissues in the infected mice [26]. Figure 3 shows a representative image of these experiments. The heart tissues were hybridized with a *B. burgdorferi-*specific 16S rDNA probe to localize *B. burgdorferi* DNA (Figure 3A) and with the biofilm marker alginate (Figure 3B). The morphology of all the tissues was visualized using differential interference contrast microscopy (DIC) which showed the structure of the biofilm and how it is embedded in the tissue (Figure 3C). To demonstrate the specificity of the 16S rDNA probe, three different negative control conditions were employed: competing oligo (Figure 3D), DNase 1 treatment (Figure 3E), and random probe (Figure 3F) as described in Methods. None of the negative controls showed any detectable signal demonstrating the specificity of the FISH method.

### 3.3. Three-Dimensional (3D) Microscopic Analysis of B. burgdorferi Biofilm in Infected Mouse Heart Tissues

A *B. burgdorferi* positive mouse heart tissue section was stained for *B. burgdorferi* (green) and alginate (red) with DAPI nuclear stains (blue) and imaged using a wide-field fluorescent microscope (Thunder, Leica Microsystems, Wetzlar, Germany) capable of making 3D images and Z-stacks. Figure 4 shows a 3D view and a Z-stack of the embedded *B. burgdorferi* biofilms with surrounding spirochetes in an infected mouse heart tissue offering an enhanced perspective of the spatial arrangement and relationships between the various components within the tissue.

### 3.4. Ultra-Structural Analysis of the B. burgdorferi and Alginate Positive Aggregates in Mouse Heart Tissues by Atomic Force Microscopy (AFM)

In the next set of experiments, the ultrastructural organization of *B. burgdorferi* and alginate positive aggregates were further analyzed by atomic force microscopy (AFM). Figure 5 shows representative micrographs of a positively stained *B. burgdorferi* aggregate co-stained with both *B. burgdorferi*- and alginate-specific antibodies (Panels B and C respectively), which is deeply embedded in the surrounding tissues as depicted by DIC microscopy (Panel D). AFM topographical scans (Panel A) confirmed that the aggregates are indeed embedded in the tissues (indicated by the arrows) and have the biofilm-specific characteristic channels and protrusions as described before for *B. burgdorferi* biofilm structures [14,37].

### 3.5. B. burgdorferi Biofilm Effect on Host Inflammatory Response in Mouse Heart Tissues

To assess inflammatory markers near *B. burgdorferi* spirochetes and biofilms in infected mouse heart tissues, we randomly selected ten *B. burgdorferi*-positive tissue sections. These sections were immunostained with a C-reactive protein (CRP) antibody, a well-established marker for carditis [38]. Figure 6 provides a representative image for *B. burgdorferi* (green staining, Panel B) and CRP-positive tissues (red staining, Panel C). The tissue morphology was visualized using DIC revealing structural details (Panels D, H). Nuclear DNA was stained with DAPI (blue color, Panels A, E). Uninfected mouse heart tissues did not exhibit staining for either *B. burgdorferi* or CRP. (Figure 6: Panels F, G respectively). The infected mouse heart tissues had very significant CRP expression in the vicinity of *B. burgdorferi* spirochetes and biofilms.

## 4. Discussion

*Borrelia burgdorferi* has been shown to form a highly antibiotic-tolerant form called biofilm, reported in vitro and in vivo [14,23,25,26]. It has also been demonstrated that *B. burgdorferi* can persist in the human body even after long-term antibiotic treatment not only in the spirochetal but also in the antibiotic-tolerant biofilm form [26]. This study aimed to explore whether mice could serve as a model organism to investigate the relationship between *B. burgdorferi* biofilms and their impact on host tissues. The hypothesis proposed that *B. burgdorferi* could form biofilms in mouse heart tissues potentially leading to inflammation and damage in the surrounding infected host tissue. Here, we utilized a well-established C3H/HeN mouse infection model system [31,32,33,34] to confirm the presence of *Borrelia burgdorferi* biofilms. Initially, successful infection and the manifestation of Lyme disease and arthritis were validated through culture, ankle joint measurement, qPCR, and ELISA methods. Multiple methods were used to verify that the aggregates observed in the mouse tissues are indeed *B. burgdorferi* biofilms and not biofilms from another species. To specifically identify *B. burgdorferi*, we employed fluorescent immunohistochemistry (IHC) and fluorescence in situ hybridization combined with IHC (FISH/IHC) techniques on mouse tissue samples. These techniques revealed the presence of *B. burgdorferi* -positive spirochetes and aggregates positive for both *B. burgdorferi* and alginate in infected mouse heart tissues.

Our findings align with a clinical case study of a 53-year-old female Lyme disease patient, in which *B. burgdorferi* spirochetes and biofilms were also detected in heart autopsy tissues [26]. The consistency between our experimental results and the human study highlights the relevance and clinical significance of our findings.

To validate the specificity of our IHC experiments, we utilized additional uninfected mouse heart tissue samples as negative controls which were subjected to the same IHC staining method as mentioned above and no evidence was found that these tissues stained for either *B. burgdorferi* or alginate antigens. We also included a non-specific IgG primary antibody as part of our controls which further confirmed that our primary antibodies are specific for their targets.

To provide additional evidence for *B. burgdorferi* biofilms in infected mouse heart tissues, we employed a previously published dual FISH/HIC protocol [23]. This technique included several rigorous negative controls: random probe, competing oligo, and DNase 1 treatment to ensure the specificity of the FISH results. The 3D microscopy imaging provided valuable insights into the spatial distribution and co-localization of *B. burgdorferi* spirochetes and biofilms within the mouse heart tissue. The combination of specific staining and advanced imaging capabilities enabled the visualization of *B. burgdorferi* biofilm within the heart tissue, shedding light on how biofilm is embedded in the tissue. Ultrastructural analysis using atomic force microscopy (AFM) revealed the intricate organization of *B. burgdorferi* and alginate dual-positive aggregates within the infected mouse heart tissues. AFM scans confirmed the presence of these aggregates and their characteristic channels and protrusions, providing further evidence of biofilm formation. These observations are consistent with findings from earlier studies [14,37] and similar aggregates identified in borrelial lymphocytoma (BL) biopsy tissues [23].

Previous research conducted on *B. burgdorferi*-infected mice showed the colonization of disseminating spirochetes in the heart and joints resulting in acute inflammation with carditis and arthritis [31,32,33,34]. Further examinations of infected mouse heart and joint tissues after 30, 90, 180, and 360 days showed persistent spirochetes in collagenous tissues with inflammation suggesting recurrent acute diseases [32,33]. Another study visualized the presence of spirochetes within cardiac fibrosis in the early phase of infection [34] and also showed the presence of spirochetes in the aortic wall at the basal membrane of the heart [38,39]. A study done on *Rhesus macaques* suggested that *B. burgdorferi* cannot be eliminated by standard doxycycline treatment in the heart tissues and can be detected by IHC and PCR technology but were unculturable [40]. Studies related to non-cultivable *B. burgdorferi* in other animal models like mice have also shown RNA transcription of targeted genes from *B. burgdorferi* post-antibiotic treatment for up to 12 months [7,40]. Another study using a canine model showed the presence of persisting *B. burgdorferi* DNA in various tissues, despite antibiotic treatment [41,42,43]. Taken together, these model studies suggest that non-cultivable *B. burgdorferi* may become tolerant to environmental changes, but further studies using reproducible animal models are warranted to more closely examine the question of persistence.

As previously mentioned in the context of Lyme disease, biofilm formation by *B. burgdorferi* can provide a refuge for the bacteria making them tolerant to antibiotics and unfavorable environmental changes leading to chronic infection. Persistent infection caused by *Borrelia burgdorferi* triggers an ongoing inflammatory response [34]. Biofilms have been implicated in several chronic infections caused by different pathogens. Examples of such infections include *Pseudomonas aeruginosa* infections associated with cystic fibrosis, keratitis resulting from contact lenses, urinary tract infections from *Escherichia coli*, osteomyelitis or endocarditis by *Staphylococcus aureus*, and pulmonary infections from *Streptococcus pneumoniae* [44,45,46,47]. This resilience can result in persistent infections and recurrent episodes of inflammation and tissue damage [48].

Various inflammatory markers are utilized to detect chronic inflammation following *B. burgdorferi* infection, with elevated C-reactive protein (CRP) being particularly significant [49]. CRP is an acute-phase protein that serves as a biomarker for inflammation in the body. It is known to recognize foreign pathogens and damaged cell phospholipids, and its synthesis is primarily induced by the liver during inflammatory responses [50,51,52]. The known association of CRP with periodontal biofilms and its presence in human heart tissues during myocardial infection further underscores its role in recognizing and responding to microbial infections [53]. In our previous research, elevated levels of C-reactive protein were observed in *B. burgdorferi* biofilm-positive human heart autopsy tissues, consistent with our findings in mouse heart tissues [54]. Several studies also provided strong evidence linking *B. burgdorferi* to inflammation and the pathogenesis of Lyme carditis [55,56,57]. These findings collectively highlight CRP as a crucial marker in understanding the inflammatory response induced by *B. burgdorferi* and its implications in Lyme disease progression.

## 5. Conclusions

In summary, our findings provide evidence that a murine model can be used to study the association between *B. burgdorferi* biofilm formation and the development of Lyme carditis. Given its potential severity and implications for patient health, studying Lyme carditis is crucial for advancing our understanding of the disease and improving patient outcomes.

## Figures and Tables

**Figure 1 microorganisms-12-01766-f001:**
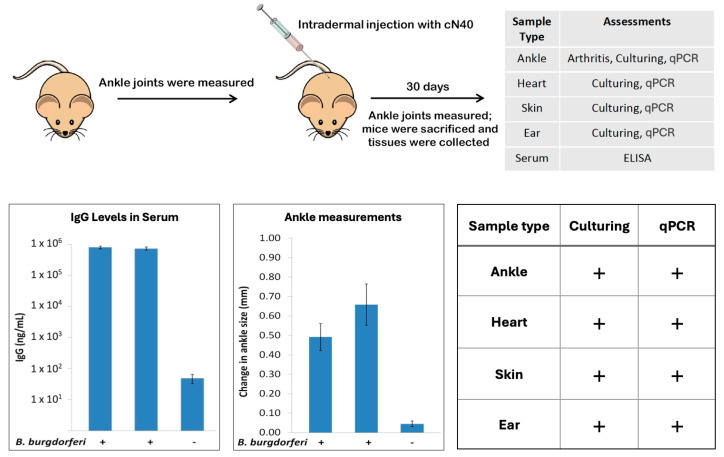
Summary of the design of the experimental infection of C3H/HeN mice with *B. burgdorferi* and the collection of different samples (upper panel). The lower left panel shows the results of ELISA for IgG serum levels in the two infected groups (20 mice/group) and one uninfected group (4 mice). The lower middle panel shows results of the ankle measurements in the infected and uninfected groups, while the lower right panel summarizes the results of the culture and qPCR experiments on infected mice.

**Figure 2 microorganisms-12-01766-f002:**
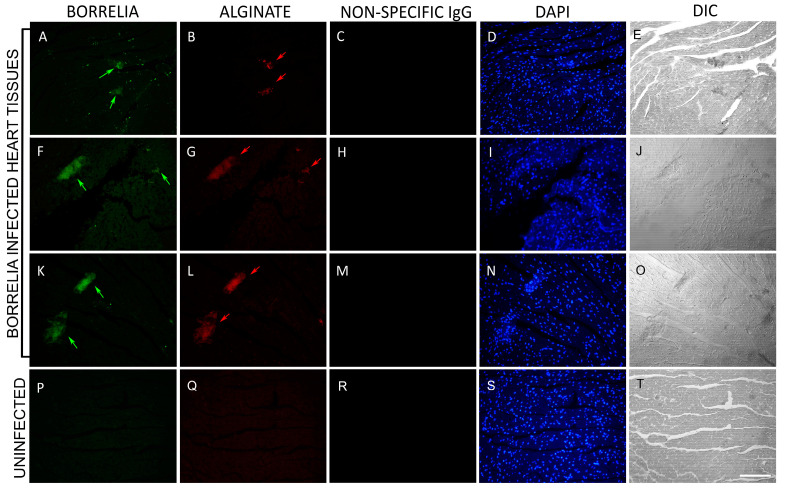
Immunohistochemical (IHC) detection of *B. burgdorferi* spirochetes and biofilm in heart tissue sections of C3H/HeN infected and uninfected mice. Panels (**A**,**F**,**K**,**P**) show IHC results using a FITC-labeled anti-borrelia antibody. Green arrows depict *B. burgdorferi* aggregates; green arrowheads point to small spirochetes. Panels (**B**,**G**,**L)** show IHC results using an anti-alginate antibody (red staining). Red arrows depict the presence of alginate on *B. burgdorferi* aggregates. Panels (**C**,**H**,**M**,**R**) show IHC results with non-specific IgG antibody (negative control). Panels (**D**,**I**,**N,S**) show DAPI stain for nuclear DNA. Panels (**E**,**J**,**O**,**T**) show the structure of the tissues by DIC microscopy. Panels (**P**,**Q**) show uninfected mouse heart sections that were stained with the same *B. burgdorferi* and alginate antibodies. Images were taken at 400×. Bar: 200 μm.

**Figure 3 microorganisms-12-01766-f003:**
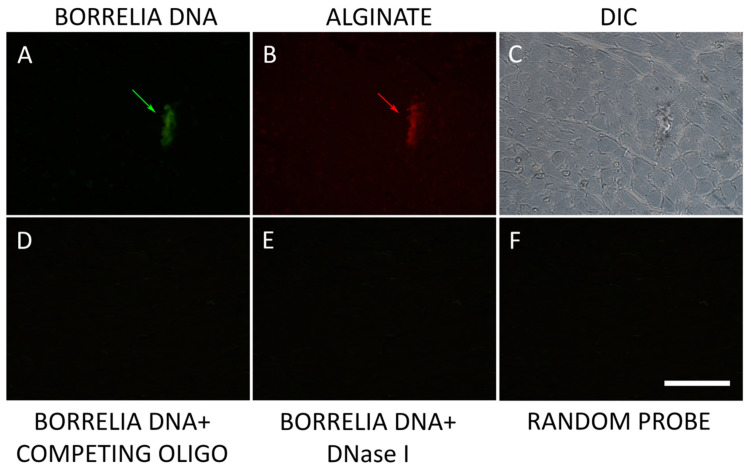
Representative images of *B. burgdorferi and* alginate-specific combined FISH/IHC on infected mouse heart tissue. Panel (**A**) shows *B. burgdorferi* stained with fluorescently labeled 16S rDNA probe (green staining with a green arrow). Panel (**B**) shows IHC results using an anti-alginate antibody on subsequent section (red staining with a red arrow). Panel (**C**) shows the structure of the tissue (DIC). Panels (**D**–**F**) show negative controls for the FISH experiments: competing oligo, DNase 1, and random probes respectively as described in Methods. Images were taken at 400×. Bar: 200 μm.

**Figure 4 microorganisms-12-01766-f004:**
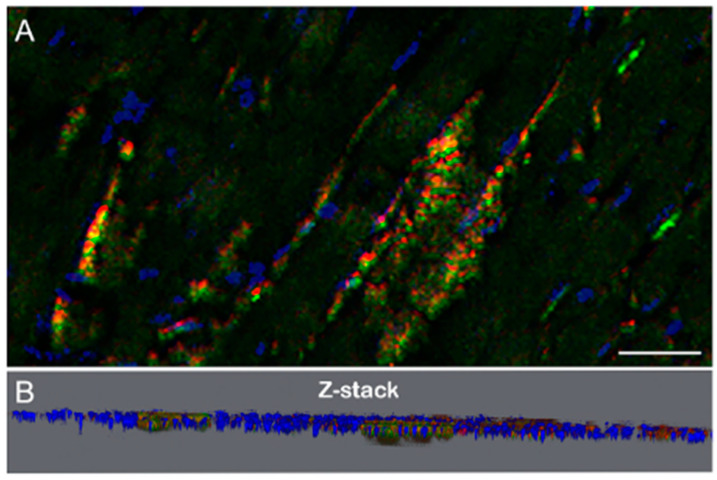
Representative IHC images of *B. burgdorferi* infected mouse heart tissue section demonstrating the 3D spatial arrangement (Panel (**A**)) with the corresponding Z-stack (Panel (**B**)). Green staining: *B. burgdorferi*, red staining: alginate and blue staining: DAPI. Bar: 20 μm.

**Figure 5 microorganisms-12-01766-f005:**
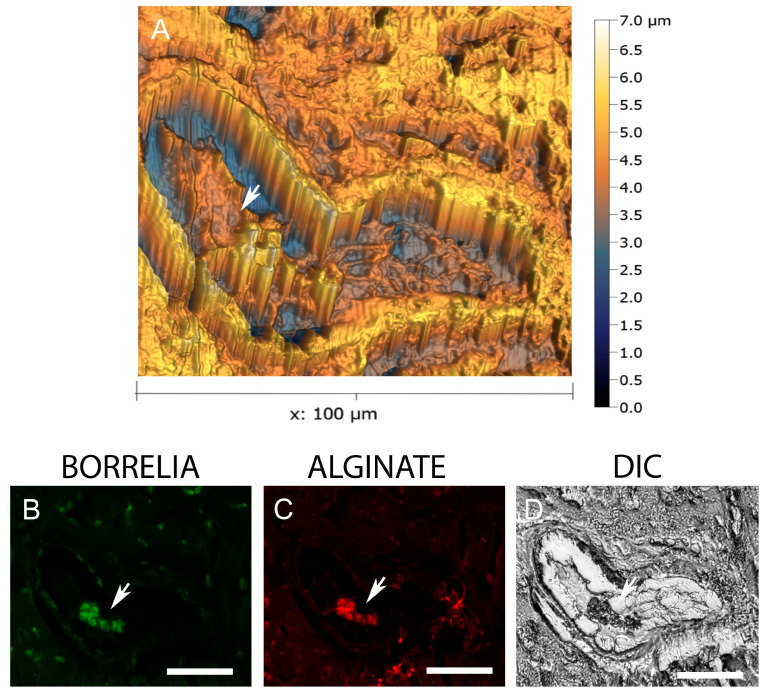
Three-dimensional analysis of Borrelia aggregates stained positive for *B. burgdorferi* and alginate via atomic force microscopy (AFM) in infected mouse heart tissue sections. Panel (**A**) shows the AFM topographical scans of *B. burgdorferi* biofilm (white arrowhead). Panel (**B**) shows results using a FITC-labeled anti-*Borrelia* antibody (green staining, white arrowhead). Panel (**C**) shows results using an anti-alginate antibody (red staining, white arrowhead). Panel (**D**) shows the DIC image of the infected mouse tissue section (white arrowhead), which was used for the AFM study. Scale bar: 50 μm.

**Figure 6 microorganisms-12-01766-f006:**
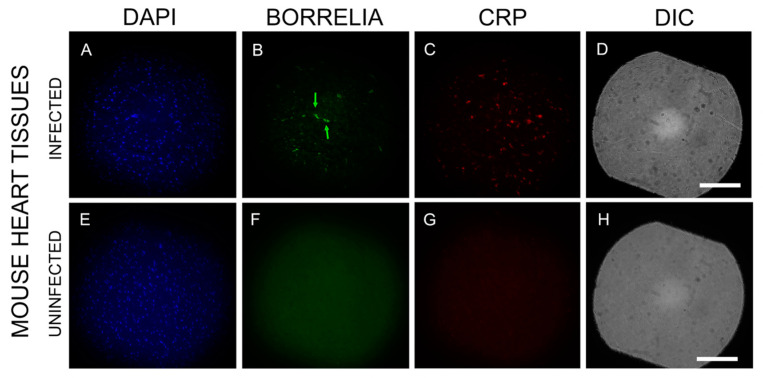
Representative IHC images of *B. burgdorferi* and C-reactive protein (CRP) staining in infected and uninfected C3H/HeN mouse heart sections. Panels (**B**,**F**) show results using a *B. burgdorferi*-specific antibody (green)**.** Panels (**C**,**G**) show results using a CRP antibody (red). Panels (**A**,**E**) show DAPI stain for nuclear DNA. Panels (**F**,**G**) show negative control, uninfected mouse heart sections that were stained with the same *B. burgdorferi* and CRP antibodies. Panels (**D**,**H**) show the structure of the tissues by DIC microscopy. Green arrows indicate two small *B. burgdorferi* aggregates. Images were taken at 200×. Bar: 200 μm.

## Data Availability

The original contributions presented in the study are included in the article/Appendix A, further inquiries can be directed to the corresponding author.

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
