# Peer review of "Evidence for the Presence of Borrelia burgdorferi Biofilm in Infected Mouse Heart Tissues"

_microorganisms, 2024, doi:10.3390/microorganisms12091766_

Round 1

Reviewer 1 Report

Comments and Suggestions for Authors

After reading the original article entitled “Evidence for the presence of Borrelia burgdorferi biofilm in infected mouse heart tissues”, I believe that the main aim of the article is very interesting and worth investigation. This is even more important in case of microbes, which are difficult-to-culture or are just non-culturable. Therefore, I think that this fact should be appreciated. Regarding some improvements, below I would like to present a short list:

-          In all places of the manuscript I would consider changing the terms “antimicrobial-resistant biofilms”, “antibiotic-resistant form called biofilm” or all similar in the meaning to antimicrobial-tolerant, antibiotic-tolerant called biofilm. Resistance is generated by genetic mutations, while mostly insensitivity of microbes in the biofilm is connected with “passive” mechanisms, e.g., presence of thick matrix or a slow growth rate -> therefore, the term “tolerant” is more correct.

-          Lines 28-29: “with a known biofilm marker” -> with a known biofilm marker (alginate) [I think it is worth to be added here]

-          Lines 43, 75, 83, 275, 329, 376, 377: “Borrelia” -> should be written using italics

-          Lines 46-47: “ “round bodies (cysts and granules) ” -> “round bodies (cysts and granules)”

-          Line 50: Additional space discovered, please delete

For the future: It would be interesting to: a) use the model to check activity of antibiotics against such biofilm forms developed in vivo; b) consider to check if such biofilm forms are also build up from host-derived components, e.g., fibrin, collagen, and others (as they may limit the access of antibiotics to the biofilm formed by Borrelia)

Author Response

We would like to thank our reviewer for the detailed and constructive review of our manuscript. We have made every attempt to address all concerns and suggestions and make this manuscript ready for publication.

After reading the original article entitled “Evidence for the presence of Borrelia burgdorferi biofilm in infected mouse heart tissues”, I believe that the main aim of the article is very interesting and worth investigation. This is even more important in case of microbes, which are difficult-to-culture or are just non-culturable. Therefore, I think that this fact should be appreciated. Regarding some improvements, below I would like to present a short list:

-          In all places of the manuscript I would consider changing the terms “antimicrobial-resistant biofilms”, “antibiotic-resistant form called biofilm” or all similar in the meaning to antimicrobial-tolerant, antibiotic-tolerant called biofilm. Resistance is generated by genetic mutations, while mostly insensitivity of microbes in the biofilm is connected with “passive” mechanisms, e.g., presence of thick matrix or a slow growth rate -> therefore, the term “tolerant” is more correct.

-          Lines 28-29: “with a known biofilm marker” -> with a known biofilm marker (alginate) [I think it is worth to be added here]

-          Lines 43, 75, 83, 275, 329, 376, 377: “Borrelia” -> should be written using italics

-          Lines 46-47: “ “round bodies (cysts and granules) ” -> “round bodies (cysts and granules)”

-          Line 50: Additional space discovered, please delete

For the future: It would be interesting to: a) use the model to check activity of antibiotics against such biofilm forms developed in vivo; b) consider to check if such biofilm forms are also build up from host-derived components, e.g., fibrin, collagen, and others (as they may limit the access of antibiotics to the biofilm formed by Borrelia)

Thank you for your positive feedback on our manuscript and for providing detailed comments on the issues you identified. Your help with this manuscript is greatly appreciated. We have agreed and addressed all 5 suggested changes and have marked the modifications in red in the revised manuscript. Thank you also for the great suggestions for future studies.

Reviewer 2 Report

Comments and Suggestions for Authors

The authors mentioned “In our previous studies, we provided both in vitro and in vivo evidence that B. 60 burgdorferi biofilm exists within infected human tissues”. It would be interesting to compare the biofilm for human tissues and for mouse heart tissues

The authors mentioned alginate. The paper would be strengthened if more biofilm matrix components were quantified.

While DNA extraction was proposed, other bacteria may form biofilms. Discussion is needed for this.

Is there a way to quantify the observation shown in Figure 2? In addition, nothing is observed for the column of non-specific IgG.

Similarly, nothing is observed in the second row in Figure 3

The authors mentioned that there are 20 mice total in 2 111 groups. Not sure how consistency was observed for Figures 2 & 3 for these mice.

There is no statistical analysis of the results, no statistical comparison between control and infected groups. No evidence for the statistical significance of the findings from this project.

Comments on the Quality of English Language

The English is generally fine. 

Author Response

We would like to thank our reviewer for the detailed and constructive review of our manuscript. We have made every attempt to address all concerns and suggestions and make this manuscript ready for publication.

The authors mentioned “In our previous studies, we provided both in vitro and in vivo evidence that B. 60 burgdorferi biofilm exists within infected human tissues”. It would be interesting to compare the biofilm for human tissues and for mouse heart tissues.

Thank you for the suggestions. We plan to compare Borrelia burgdorferi-infected tissues from humans and mice, with a particular focus on common inflammatory markers. In our previous studies, we identified the CXCL and CCL chemokine families as key inflammatory markers in B. burgdorferi-infected cells. For this study, we investigated several inflammatory markers, including C-reactive protein (CRP), CXCL8 and CXCL10, in relation to biofilm structures in infected mouse heart tissues. However, we have included only data on CRP staining in this manuscript because it yielded the strongest and most specific signal, consistent with our findings from a human autopsy study.3

  1. Sapi E, Kasliwala RS, Ismail H, Torres JP, Oldakowski M, Markland S, Gaur G, Melillo A, Eisendle K, Liegner KB, Libien J, Goldman JE. (2019) The long-term persistence of Borrelia burgdorferi antigens and DNA in the tissues of a patient with Lyme Disease. Antibiotics, 8(4):183. Doi: 10.3390/antibiotics8040183
  2. Khatri VA, Paul S, Patel NJ, Thippani S, Sawant JY, Durkee KL, Cassandra L. Murphy CL, Ortiz GA, Valentino JA, Jathan J, Anthony Melillo M, Sapi E. (2023) Global transcriptomic analysis of breast cancer and normal mammary epithelial cells infected with Borrelia burgdorferi. European Journal of Microbiology and Immunology. 13(3): 63-76.
  3. Kasliwala R. Effect of Borrelia biofilm on the host inflammatory markers in infected human tissues. [Master’s thesis]. West Haven, CT: University of New Haven; 2017).

  1. The authors mentioned alginate. The paper would be strengthened if more biofilm matrix components were quantified.

We agree that incorporating additional biofilm markers from our previous study4, such as HHA and MOA lectins and Poly-N-acetylglucosamine, would have been advantageous for examining Borrelia burgdorferi biofilm. However, these matrix markers resulted in high background noise in immunohistochemistry (IHC) on mouse tissues, making them unsuitable for our study. Instead, we used alginate, a well-characterized mucopolysaccharide associated with biofilms. The antibody we employed, developed by Dr. Gerald Pier from Harvard University, produced no background signal on either human or mouse tissues.

  1. Shaikh S, Timmaraju VA, Torres JP, Socarras KM, PAS, Sapi E. (2016) Influence of tick and mammalian physiological temperatures on Borrelia burgdorferi Microbiology, 162: 1984-1995,doi: 10.1099/mic.0.000380

While DNA extraction was proposed, other bacteria may form biofilms. Discussion is needed for this.

Yes, we acknowledge that other bacteria can form biofilms. To confirm that the biofilms we identified were indeed B. burgdorferi biofilms, we employed two different techniques—immunohistochemistry (IHC) and fluorescence in situ hybridization (FISH)—along with several negative controls. We included a statement in the discussion to explain that multiple methods were used to verify that the aggregates observed are B. burgdorferi biofilms (marked in red in the revised manuscript, lines 373-375).

Is there a way to quantify the observation shown in Figure 2? In addition, nothing is observed for the column of non-specific IgG.

In the results section, we reported that “quantitative analyses showed that infected heart tissues had an average of 35 spirochetes and 0-6 biofilms per section, with biofilm sizes ranging from 20-200 micrometers.” We did not provide additional details primarily because quantifying spirochetes in tissue sections is challenging. Spirochetes are 10 micrometers long and 2 micrometers in diameter; depending on tissue sectioning, a 2-micrometer dot may not definitively indicate a spirochete. Additionally, results for non-specific IgG were completely negative, with no fluorescent signal detected. We included this antibody in our human studies as well, based on recommendations from previous reviewers who required it as an important negative control for immunohistochemistry (IHC) experiments.

Similarly, nothing is observed in the second row in Figure 3.

The second row of Figure 3 represents three different negative controls for the FISH experiments. None of the negative control produced detectable signals which proved that our FISH method is specific for B. burgdorferi biofilm. 

The authors mentioned that there are 20 mice total in 2 111 groups. Not sure how consistency was observed for Figures 2 & 3 for these mice.

Figures 2 and 3 are representative of our experiments. Additional IHC figures showing more biofilms in these tissues are provided in the supplementary file. The IHC and FISH data were obtained from 200 sections across 20 heart samples. It is important to note that heart tissues were also required for other experiments, including culture and qPCR analyses. We had to use some of the archived tissues to troubleshoot and optimize our IHC and FISH techniques. Additionally, we stained for various biofilm and inflammatory markers, which limited the available tissue and necessitated careful distribution among the different experiments.

There is no statistical analysis of the results, no statistical comparison between control and infected groups. No evidence for the statistical significance of the findings from this project.

Our study did not aim to conduct a comprehensive quantitative analysis for several reasons. The primary objective was to provide evidence that B. burgdorferi biofilms can form in heart tissues, a phenomenon not previously demonstrated. We concentrated on visualizing and characterizing these biofilms using various microscopy techniques, including atomic force microscopy. Additionally, we faced limitations in tissue availability despite having 20 infected mice. A significant portion of the tissues was needed for culture and qPCR experiments to ensure successful infection. We plan to conduct a larger study with more mice in the future to enable quantitative analyses and to explore other organs and markers.

.